# Assessing AMOC stability using a Bayesian nested time-dependent autoregressive model

Luc Hallali<sup>1</sup>, Eirik Myrvoll-Nilsen<sup>1</sup>, and Christian L. E. Franzke<sup>2,3</sup>

<sup>1</sup>Department of Mathematics and Statistics, UiT The Arctic University of Norway, N-9037 Tromsø, Norway

<sup>2</sup>Center for Climate Physics, Institute for Basic Science, Busan, Republic of Korea

<sup>3</sup>Department of Integrated Climate System Science, Pusan National University, Busan, Republic of Korea

Correspondence: Luc Hallali (luc.hallali@uit.no)

# Abstract.

The Atlantic Meridional Overturning Circulation (AMOC) is a major climate element subject to possible ongoing loss of stability. Recent studies have found evidence of a gradual weakening in circulation, including early warning signals (EWS), such as increased fluctuations and correlation time of the system, which are both known to be indicators of a possible forth-coming tipping point. To assess these changes in statistical behavior we propose a robust and general statistical model based on a second-order autoregressive process with time-dependent parameters. This allows for the statistical changes from increased external variability and destabilization to be accounted for separately. We estimate the time evolution of the correlation parameters using a hierarchical Bayesian modeling framework which also yields uncertainty quantification through the posterior distribution. To assess possible changes in AMOC stability we apply the model to an AMOC fingerprint proxy based on the Sub-Polar Gyre and the global mean temperature anomaly. We find statistically significant EWS which suggests that AMOC is indeed undergoing a loss of stability and is getting closer to a tipping point. The methodology developed in this study is made publicly available as an extension of the R-package INLA, ews.

# 1 Introduction

The Atlantic Meridional Overturning Circulation (AMOC) is a key driver of Earth's climate, responsible for the transport of heat and salt across the Atlantic Ocean (Rahmstorf, 1995). As part of the global thermohaline circulation, the AMOC plays a central role in maintaining the current climate equilibrium. It is widely believed that the AMOC is a multi-stable system, capable of existing in multiple stable modes, most notably a strong mode, which is currently dominant, and a weak or collapsed mode (Stommel, 1961; Lenton et al., 2008). This nonlinear behavior implies that the AMOC may undergo abrupt transitions between states when critical thresholds are crossed. Paleoclimate evidence supports the idea that abrupt shifts in AMOC strength have contributed to major climate events during the last glacial period, such as the Dansgaard-Oeschger events (Vettoretti and Peltier, 2016; Boers et al., 2018).

These dynamics have led to the identification of the AMOC as a potential "tipping element" in the Earth system, i.e., a subsystem that could undergo a critical transition due to anthropogenic forcing (Lenton et al., 2008). Climate models suggest that continued greenhouse gas emissions and the resulting increase in freshwater input from Greenland Ice Sheet melt, precip-

itation, and river discharge could push the AMOC toward such a tipping point (Wood et al., 2019; Hawkins et al., 2011; Weijer et al., 2019). This behavior exhibits hysteresis, meaning that once a tipping threshold is passed, the AMOC may not return to its original state even if the perturbation is reversed.

Recent observational and modeling studies have intensified concerns. Although early models suggested a low probability of collapse within the 21st century (Masson-Delmotte et al., 2021), more recent simulations reveal a wider range of possible responses, raising concerns that risks might be underestimated (Gong et al., 2022). This discrepancy is partly due to model biases, notably in representing freshwater forcing and feedback (Liu et al., 2017). Evidence is also emerging from real-world observations. Studies have documented a significant weakening trend in the AMOC over the 20th century (Caesar et al., 2018) and recent statistical analyses have detected early warning signals of reduced stability (Boers, 2021; Ditlevsen and Ditlevsen, 2023). These findings suggest that the AMOC may be approaching a critical threshold.

A weakening of the AMOC would have profound and potentially irreversible consequences, including disrupting weather patterns, altering precipitation systems, and potentially triggering cascading effects on other climate components (Stouffer et al., 2006; Jackson et al., 2015; Lenton et al., 2008). In light of this, there is an urgent need to monitor the resilience of the system and improve our understanding of the processes that drive its potential loss of stability. To anticipate such changes studies have focused on using critical slowing down theory, stating that when a system is approaching a tipping point its recovery from small perturbations becomes progressively weaker. This phenomenon, called early-warning signal (EWS), can be characterized by an increased variance and autocorrelation which can be used as statistical indicators of approaching critical transitions.

To detect these statistical changes, a common approach is to start from the linear approximation of a dynamical system with white noise around some stable fixed point  $x_s$ , giving

45 
$$dx(t) = -\lambda(x(t) - x_s)dt + \sigma dB(t), \tag{1}$$

where  $\lambda$  is the restoring rate and dB(t) is a white noise process. This Linearization is recognized as the Langevin stochastic differential equation, which has the following solution

$$x(t) = x_0 + \int_{-\infty}^{t} g(t-s)dB(s), \tag{2}$$

where g(t-s) is a Green's function defined by


$$g(t) = \begin{cases} \exp(-\lambda t), & x \ge 0 \\ 0, & x < 0 \end{cases}$$
 (3)

This form of x(t) is also referred to as an Ornstein-Uhlenbeck (OU) process. When discretized, this process yields a first-order autoregressive (AR) process.

$$x_t = \phi x_{t-1} + \varepsilon_t, \qquad \varepsilon_t \sim \mathcal{N}\left(0, \frac{1 - \phi^2}{2\lambda}\sigma^2\right)$$
 (4)

with variance  $Var(x_t) = \sigma^2/(2\lambda)$  and lag-one autocorrelation parameter  $\phi = \exp(-\lambda \Delta t)$ .



With this model, EWS are detected through an increase of the autocorrelation or variance. However, Boers (2021) showed that these indicators can be biased if the system is driven by external noise that itself has increasing autocorrelation or variance, leading to false positive alarms. To account for such bias, Boettner and Boers (2022), and Morr and Boers (2024) suggests that the OU process of Eq. (2) should be driven by correlated noise rather than white noise. After discretization the resulting process yields an AR(1) process that is driven by another AR(1) process. Hence the discretization is similar to Eq. (4), except that the white noise process  $\varepsilon_t$  is replaced by an AR(1) process

$$v_{t+1} = \rho v_t + \sigma_v \xi_t \tag{5}$$

with  $\rho$  representing the correlation parameter of the noise,  $\sigma_v$  is a scaling parameter and

$$\xi_t \sim \mathcal{N}\left(0, \frac{1 - \phi^2}{2\lambda}\right) \tag{6}$$

is a white noise process. This model encompasses the original AR(1) model in (4) when  $\rho = 0$  and, as showcased in Boers (2021), it comprehends cases in which external noise is also correlated, preventing bias in the estimation of the parameter  $\phi$ . Consequently, an increasing  $\phi$  will act as a more reliable indicator for detecting EWS, since it will no longer be affected by rising external variation.

Climate systems that are prone to tipping, such as the AMOC, are often driven by some external forcing. For the AMOC, the freshwater forcing from Greenland melts acts like a bifurcation parameter as freshwater inputs can disturb the salinity and the temperature of the AMOC, potentially pushing the system closer to its tipping (Wood et al., 2019). To incorporate forcing into our model, we use a similar approach as in Myrvoll-Nilsen et al. (2024), and Myrvoll-Nilsen et al. (2020), where the dynamical system is represented by

$$dx(t) = -\lambda x(t) + F(t)dt + U(t)dt, \tag{7}$$

where F(t) represent the forcing and U(t), as before, represents an OU process. The solution of this equation can be expressed as the sum of one forced component and one noise component

$$x(t) = \nu(t) + \xi(t). \tag{8}$$

Here, the noise component,  $\xi(t)$ , is represented by an nested AR(1) process described previously and the forced component,  $\nu(t)$ , is expressed by

$$\nu(t) = \frac{1}{\sqrt{2\lambda(t)\kappa_f}} \int_0^t F(s)e^{-\lambda(t)(t-s)} ds. \tag{9}$$

with  $\kappa_f$  being a scaling parameter. This model allows EWS to be detected while accounting for the influence of external forcing on the system's dynamics.

Most studies detect EWS using sliding windows to obtain estimates of the variance and correlation for each window. This approach requires selecting an appropriate window length, which introduces a fundamental compromise. A shorter window provides a more accurate representation of the system's momentary state, but the limited number of data points can reduce the reliability of the statistical estimates. In contrast, a longer window improves the robustness of these estimates by incorporating more data, but it does so at the cost of responsiveness, as it averages information over a broader time scale and may fail to capture short-term fluctuations effectively. Determining the optimal window length is thus a critical but challenging task, as it should ideally balance estimation accuracy with the ability to reflect rapid changes in the system's evolution. Myrvoll-Nilsen et al. (2024) propose an alternative model-based approach that eliminates the need for this choice. Instead of relying on a fixed window length, the correlation parameter is assumed to evolve over time according to a predefined linear structure. This assumption enables a hierarchical Bayesian model formulation, enabling the use of well-established computational techniques to infer the parameters of the linear structure. Furthermore, Myrvoll-Nilsen et al. (2024) adopts a Bayesian framework which offers the additional advantage of providing uncertainty quantification in the form of posterior distributions, making the analysis more robust and interpretable.

In this paper we build upon the hierarchical Bayesian framework developed by Myrvoll-Nilsen et al. (2024) to integrate the nested AR(1) model proposed by Morr and Boers (2024) and Boettner and Boers (2022). This extension helps mitigate false-alarms caused by correlated noise and eliminates the need for sliding time windows, while benefiting from the advantages of a Bayesian modeling framework. This approach is then applied to an AMOC fingerprint in order to assess its potential loss of stability.

The paper is organized as follows. Section 2 outlines our methodology for the Bayesian modeling framework, including details on how inference can be obtained efficiently. In Section 3 we evaluate our model's accuracy and reliability on simulated data, assess the robustness to false alarms under increasing external variability, and benchmark its performance on real data against existing approaches. In Section 4, we use our Bayesian framework to identify EWS in an AMOC fingerprint, using different detrending strategies. Further discussion and conclusions are provided in Section 5.

# 105 2 Bayesian modeling

95

100

We assume that the observations,  $\boldsymbol{y} = (y_1,..,y_n)^{\top}$ , is expressed by

$$y = \mu + x \tag{10}$$

where the forcing response  $\boldsymbol{\mu} = (\mu_1, ..., \mu_n)^{\top}$  is expressed by

$$\mu_t = \sigma_f(t) \sum_{s=0}^t F(s) e^{-\lambda(t)(t-s+0.5)} ds$$
(11)

and the correlated time-dependent noise,  $\boldsymbol{x} = (x_1, ..., x_n)^{\mathsf{T}}$ , is given by a nested AR(1) process

$$x_{t+1} = \phi x_t + v_{t+1}$$

$$v_{t+1} = \rho v_t + \sigma_v \xi_t.$$
(12)

To model the evolution of the autocorrelation parameters we assume that they both change linearly in time, i.e.

$$\phi(t) = a_{\phi} + b_{\phi}t, \qquad 0 \le t \le 1,$$

$$\rho(t) = a_{\rho} + b_{\rho}t, \qquad 0 \le t \le 1.$$

$$(13)$$

These are expressed by unknown parameters  $a_{\phi}, b_{\phi}, a_{\rho}$  and  $b_{\rho}$ , which are estimated by fitting the model to observed data. Early warning signals due to critical slowing down is characterized through the evolution of  $\phi(t)$ , while potential changes in external variability is captured by the latent component  $\mathbf{v} = (v_1, ..., v_n)^{\top}$ . Separating these signals prevents false alarms as discussed by Boers (2021).

To obtain robust uncertainty estimates we adopt a Bayesian framework for parameter estimation, similar to Myrvoll-Nilsen et al. (2024). Given the hierarchical nature of the model, where y is modeled in terms of  $\mu$  and x, which are themselves governed by hyperparameters  $\theta = (a_{\phi}, b_{\phi}, a_{\rho}, b_{\rho}, \sigma_{v}, \sigma_{f})$ , a latent Gaussian model formulation provides a natural and efficient framework for Bayesian inference. Both components of the model,  $\mu$  and x, depend on the parameters  $a_{\phi}$  and  $b_{\phi}$  through  $\lambda(t) = -\log \phi(t)$ . This dependency introduces a challenge for obtaining reliable inference, as the parameters may be difficult to estimate independently. We therefore choose to model the sum  $\eta = \mu + x$  as a single component. The latent Gaussian model formulation is defined in three stages as follows.

1. The first stage defines the likelihood of the model, which is assumed to be conditionally independent given the latent components  $\mu$  and x. Since the variation of the observations y is captured by the latent component  $\eta = (\eta_1, ..., \eta_n)^{\top}$ , we model y as a Gaussian distribution with mean  $\eta$  and negligible observation noise,  $\sigma_y \approx 0$ , effectively setting  $y \approx \eta$ , i.e.

$$\pi(\boldsymbol{y} \mid \boldsymbol{\eta}, \boldsymbol{\theta}) = \prod_{k=1}^{n} \pi(y_k \mid \eta_k, \boldsymbol{\theta}) = \prod_{k=1}^{n} \frac{1}{\sqrt{2\pi}} \exp\left(-\frac{(y_k - \eta_k)^2}{2\sigma_y^2}\right). \tag{14}$$

2. The second stage defines the prior distribution for the latent field  $\eta$ , given parameters  $\theta$ . This component is assigned a multivariate Gaussian prior distribution with mean vector  $\mu$  and covariance matrix corresponding to the nested AR(1) process above with time-dependent  $\phi(t)$  and  $\rho(t)$ ,

$$\pi(\boldsymbol{\eta} \mid \boldsymbol{\theta}) = \mathcal{N}_n(\boldsymbol{\mu}, \boldsymbol{\Sigma}). \tag{15}$$

Since  $\eta$  follows a nested AR(1) process, which is equivalent to an AR(2) process (Morr and Boers, 2024), its precision matrix,  $Q = \Sigma^{-1}$ , is a sparse matrix of bandwidth 2. This property enables the use of computationally efficient algorithms that substantially reduce the overall computational cost.

3. The final stage defines the prior distributions for the model parameters




$$\pi(\boldsymbol{\theta}) = \pi(b_{\phi})\pi(a_{\phi} \mid b_{\phi})\pi(b_{\rho})\pi(a_{\rho} \mid b_{\rho})\pi(\sigma_{v})\pi(\sigma_{f}). \tag{16}$$

We assign uniform prior distributions on  $b_{\phi}$ ,  $(a_{\phi} \mid b_{\phi})$ ,  $b_{\rho}$  and  $(a_{\rho} \mid b_{\rho})$ , and gamma distributions on  $\kappa_v = 1/\sigma_v^2$  and  $\kappa_f = 1/\sigma_f^2$ . Note that since we assume that both  $0 < \phi(t) < 1$  and  $0 < \rho(t) < 1$  then the parameter space of  $a_{\phi}$  and  $a_{\rho}$  depend on the current state of  $b_{\phi}$  and  $b_{\rho}$ , respectively.

The joint posterior distribution for the parameters is given by

$$\pi(\boldsymbol{x}, \boldsymbol{v}, \boldsymbol{\theta} \mid \boldsymbol{y}) = \frac{\pi(\boldsymbol{y} \mid \boldsymbol{x}, \boldsymbol{v}, \boldsymbol{\theta}) \pi(\boldsymbol{x}, \boldsymbol{v} \mid \boldsymbol{\theta}) \pi(\boldsymbol{\theta})}{\pi(\boldsymbol{y})},$$
(17)

where  $\pi(y)$  is the marginal likelihood, or evidence, of y. In particular, we are interested in the marginal posterior distribution of  $b_{\phi}$ , which can be obtained by integrating out the other parameters,  $\theta_{-b_{\phi}}$ , and latent variables

$$\pi(b_{\phi} \mid \boldsymbol{y}) = \int \pi(\boldsymbol{\theta}, \boldsymbol{x}, \boldsymbol{v} \mid \boldsymbol{y}) d\boldsymbol{\theta}_{-b_{\phi}} d\boldsymbol{x} d\boldsymbol{v}.$$
(18)

Since solving this integral analytically is often impossible to do in practice, the common approach is to instead approximate it using sampling-based approaches like Markov chain Monte Carlo (MCMC) methods (Robert et al., 1999). However, since the precision matrix of the latent Gaussian field is sparse, we can employ a number of computationally efficient algorithms for fast Bayesian inference. Specifically, we evaluate all marginal posterior distributions using the framework of integrated nested Laplace approximations (INLA) (Rue et al., 2009, 2017), which is particularly suited for these types of models. INLA is available as an R package at www.r-inla.org and presents a computationally superior alternative to MCMC. Since our model requires specific implementation using the custom modeling framework of R-INLA we have decided to make the code available as a new feature in the user-friendly R-package INLA.ews, originally developed for the model described in Myrvoll-Nilsen et al. (2024). The nested time-dependent AR(1) model can be fitted by prompting:

A more extensive demonstration of the package can be found in Myrvoll-Nilsen et al. (2024).

# 3 Assessing model accuracy and robustness



To evaluate the accuracy and robustness of the proposed time-dependent nested AR(1) model, we perform three tests. Two tests use simulated data, one from the nested AR(1) model and one from stochastic differential equations representing dynamical systems with and without loss of stability. These tests are both based on 500 independent simulated time series of length 150, matching the length of the AMOC fingerprint time series used in Section 4. Finally, we fit our model to a real data example, the Dansgaard-Oeschger events in order to compare our model's results with existing methodologies. All tests are made using R-INLA with the prior distributions described in the previous section.

#### 3.1 Model accuracy on simulated data

For the first test, we assess whether the model can recover known parameter values when fitted to simulated data generated from the same nested time-dependent AR(1) process. For each simulation, the slope parameters  $b_{\phi}$  and  $b_{\rho}$  are independently drawn from a uniform distribution  $\mathcal{U}(-0.9,0.9)$ . Thereafter, the intercepts  $a_{\phi}$  and  $a_{\rho}$  are drawn from uniform distributions with boundaries that depend on the simulated slope parameters, ensuring that the resulting  $\phi(t)$  and  $\rho(t)$  remain within the interval (0,1) for all time steps.

We compute the root mean square error (RMSE) between the true slope values and their marginal posterior means,  $\hat{b}_{\phi}$  and  $\hat{b}_{\rho}$ . We find the RMSE to be 0.145 for  $b_{\phi}$  and 0.278 for  $b_{\rho}$ . We then assess whether the model reliably infers the sign of the slopes by comparing the marginal posterior probabilities  $P(b_{\phi} > 0 \mid \boldsymbol{y})$  and  $P(b_{\rho} > 0 \mid \boldsymbol{y})$  to the true value of the slopes. We consider the slope for  $\phi(t)$  and  $\rho(t)$  to be significantly positive if the posterior probabilities exceed the threshold  $1 - \alpha = 0.95$ , i.e.  $P(b_{\phi} > 0 \mid \boldsymbol{y}) > 0.95$  and  $P(b_{\rho} > 0 \mid \boldsymbol{y}) > 0.95$ , respectively. If an estimated  $\hat{b}_{\phi}$  is classified as positive, given the  $P(b_{\phi} > 0 \mid \boldsymbol{y}) > 0.95$  threshold, we count it as a true positive if the true slope is also positive, i.e.  $b_{\phi} > 0$ . On the other hand, if the true slope is negative,we count the estimate as a false positive. Conversely, if  $P(b_{\phi} > 0 \mid \boldsymbol{y}) \leq 0.95$  we count it as a true negative if the true slope is also negative, and as a false negative if  $b_{\phi} > 0$ . We also count the classifications based on the estimated  $\hat{b}_{\rho}$ , but these are of secondary interest. The sensitivity and specificity is computed by

Sensitivity = 
$$\frac{\text{\#True Positives}}{\text{\#True Positives} + \text{\#False Negatives}}$$
, Specificity =  $\frac{\text{\#True Negatives}}{\text{\#True Negatives} + \text{\#False Positives}}$ . (19)

For  $b_{\phi}$ , the model achieves a sensitivity of 87.7% and a specificity of 99.8%. For  $b_{\rho}$ , the sensitivity is 72.2% and the specificity 99.4%. The results from this test are summarized in Table 1 and illustrated in Fig. 1. Repeating the test with different prior distributions similar to Myrvoll-Nilsen et al. (2024) did not show significant changes, suggesting that the model is robust to the choice of prior distributions.

Figure 1. Results of the accuracy test for  $n_r=500$  simulated time series of length n=150. Panels (a) and (b) show posterior marginal mean estimated by INLA for  $\phi$  and  $\rho$ , respectively. The blue line shows the true b used in the simulation. Panels (c) and (d) show the estimated posterior probability of the slope being positive against true values of  $\phi$  and  $\rho$  respectively. The horizontal red lines separates the true positive and negative values while the horizontal one indicates the probability threshold 0.95 used here to determine statistical significance.

# 185 3.2 Robustness to false alarms under autocorrelated external variability

In the second test, we evaluate the ability of the model to reliably distinguish genuine early warning signals from changes driven solely by correlated external variability. To do so, we simulate data from two stochastic differential equations. The first represents a system approaching a tipping point, and the second remains stable but is influenced by a time-dependent autocorrelated noise. This setup follows the example in Boers (2021). The tipping process is expressed by

$$\dot{x}(t) = -x^3 + x - T + v(t),$$
 (20)

where T increases linearly from -1 to 1, and v(t) is a time-dependent AR(1) process with parameters drawn in the same way as in the first test. The non-tipping process is generated by

$$\dot{x}(t) = -5x + v(t),\tag{21}$$

with the same structure for v(t) as in Eq. (20). Each simulation is run until the tipping point is reached (for the tipping system) or for 150 time units (for the non-tipping system), resulting in time series of approximately 150 points. The same inference methodology and classification thresholds are used here, with the distinction that an early warning signal is said to be detected when  $P(b_{\phi} > 0 \mid \boldsymbol{y}) > 0.95$ . For the tipping processes the model correctly detected an EWS signal in 471 out of 500 simulations, corresponding to a sensitivity of 94.2%. For the non-tipping processes, 23 out of 500 simulations were incorrectly classified as EWS, resulting in a specificity of 95.4%. These results, presented in Table 1 and Fig. 2, indicate that the model effectively identifies true loss of stability while maintaining a low false positive ratio, even in the presence of strongly autocorrelated noise.



Figure 2. Results of the robustness test for  $n_r=500$  simulated time series of length  $n\approx 150$ . Panels (a) and (b) show posterior marginal mean estimated by INLA from the non-tipping simulations for  $b_\phi$  and  $b_\rho$  respectively plotted against the true value of  $b_\rho$  from the correlated noise. The blue line in (b) shows the true  $b_\rho$  used in the simulation. Panel (c) and (d) are similar plots for the tipping simulations. In panels (a) and (c) blue dots are associated with a statistical significance for the EWS indicator  $b_\phi$  to be positive  $P(b_\phi>0\mid \boldsymbol{y})>0.95$  while red dots mean no statistical significance  $P(b_\phi>0\mid \boldsymbol{y})

Figure 3. Results of the robustness test from the Myrvoll-Nilsen et al. (2024) model applied on the same data as Fig. 3 (a). The posterior marginal mean is plotted against the true value of  $b_{\rho}$  from the correlated noise. The blue line is a linear regression on the data, showing the drift of the estimates of  $b_{\phi}$ . Blue dots are associated with a statistical significance for the EWS indicator  $b_{\phi}$  to be positive while red dots mean no statistical significance.

Overall, these tests demonstrate that the proposed methodology reliably recovers the evolution of autocorrelation parameters, performs well in detecting EWS and is robust to prior assumptions and to structured stochastic external variability not linked to loss of stability.

| Accuracy test    |                      |                |                |                               |
|------------------|----------------------|----------------|----------------|-------------------------------|
| Estimates        | RMSE                 | Sensitivity(%) | Specificity(%) |                               |
| $\hat{b_{\phi}}$ | 0.145                | 87.8           | 99.8           |                               |
| $\hat{b_{ ho}}$  | 0.278                | 72.2           | 99.4           |                               |
| Robustness test  |                      |                |                |                               |
| Process          | $\hat{b_{ ho}}$ RMSE | Sensitivity(%) | Specificity(%) | $\langle \hat{b_\phi}  angle$ |
| Tipping          | 0.34                 | 94.2           | -              | 0.47                          |
| Non-tipping      | 0.26                 | -              | 95.4           | 0                             |

**Table 1.** Summary statistics from Fig. 1 (top) and Fig. 2 (bottom). Results from Accuracy tests on simulated time-dependent nested AR(1) processes showing Root Mean Square Error (RMSE) of the estimates of  $b_{\phi}$  and  $b_{\rho}$  given true values of simulations (blue lines in panels (a)-(b) of Fig. 1). We also show the sensitivity and specificity expressed in percentages for both parameters (bottom). Results from Robustness tests on simulated tipping and non-tipping processes. We show here the RMSE of the estimates of  $b_{\rho}$  given true simulated values (blue lines in panels (b) and (d)). Sensitivity and specificity are presented in percentages for each process.

# 3.3 Benchmarking on real data: Dansgaard-Oeschger events


Figure 4. The 17 most recent Dansgaard-Oeschger (DO) events (vertical black lines) in the NGRIP  $\delta^{18}$ O record plotted against the GICC05 chronology. Early warning signals are estimated by fitting the model to the Greenland stadial periods (black segments) of the data preceding each DO event.

Finally, we evaluate the performance of our model on real-world data associated with well-studied critical transitions. By comparing results, we can assess how much our model agrees or disagrees with existing approaches. Specifically, we will use our model to analyze abrupt Greenland warmings known as Dansgaard-Oeschger (DO) events (Dansgaard et al., 1993; Johnsen et al., 1992), which represent rapid climate fluctuations that occurred during the last glacial period, where the temperature over Greenland and the North Atlantic region increased by up to  $16.5^{\circ}$ C within a few decades (Kindler et al., 2014). DO events are

often considered the archetypal example of tipping point crossings in the climate. As such, they present a natural benchmark for evaluating different EWS approaches.

For our analysis, we pair the  $\delta^{18}$ O proxy data from the Northern Greenland Ice Core Project (NGRIP) (North Greenland Ice Core Project members, 2004; Gkinis et al., 2014; Ruth et al., 2003) with the corresponding age provided by the Greenland Ice Core Project 2005 (GICC05) (Vinther et al., 2006; Rasmussen et al., 2006; Andersen et al., 2006; Svensson et al., 2008). The data is available at https://www.iceandclimate.nbi.ku.dk/data (last accessed: August 5, 2025). The model is fitted to segments preceding the 17 most recent DO event. The selected segments are highlighted in Fig. 4.




Whether or not DO events are induced solely by noise, or if they are indeed approaching a bifurcation point, is currently debated (Ditlevsen et al., 2007; Hummel et al., 2024). There is therefore no ground truth as to which, if any, DO event should exhibit EWS. However, several studies report a detection of EWS before some of the first 17 DO events (Rypdal, 2016; Boers, 2018). We compare the results of our model with these studies using a setup similar to Myrvoll-Nilsen et al. (2024) by using a second-order polynomial detrending of the data and considering  $P(b_{\phi} > 0 \mid y) > 0.95$  as a detection of EWS. This comparison is illustrated in Table 2 and shows that our model suggests, similarly to Myrvoll-Nilsen et al. (2024), that some specific event shows signs of critical slowing down in line with the results of Boers (2018) and Rypdal (2016). Specifically, Table 2 shows that our results corroborate the five EWS found by Myrvoll-Nilsen et al. (2024) while identifying one more EWS for the 13th event. Moreover, these results corroborate the EWS found for the 11th event by Boers (2018) and the 5th and 9th events by Rypdal (2016), our results also show EWS for the 2nd and 13th events similarly to these two studies.

| Event | Nested AR(1) | Myrvoll-Nilsen | Rypdal    | Boers    |
|-------|--------------|----------------|-----------|----------|
| 1     | 0.893        | 0.9146         | p = 0.02  | _        |
| 2     | 0.992        | 0.9728         | p = 0.008 | p < 0.05 |
| 3     | 0.29         | 0.4893         | _         | _        |
| 4     | 0.053        | 0.084          | _         | p < 0.05 |
| 5     | 0.99         | 0.9959         | p = 0.13  | _        |
| 6     | 0.163        | 0.2123         | _         | p < 0.05 |
| 7     | 0.444        | 0.7132         | _         | _        |
| 8     | 0.817        | 0.8878         | _         | _        |
| 9     | 0.994        | 0.953          | p = 0.16  | _        |
| 10    | 0.115        | 0.0732         | _         | _        |
| 11    | 0.977        | 0.9643         | _         | p < 0.05 |
| 12    | 0.056        | 0.1662         | _         | _        |
| 13    | 0.978        | 0.8912         | p = 0.39  | p < 0.05 |
| 14    | 0.722        | 0.6629         | _         | p < 0.05 |
| 15    | 0.061        | 0.0637         | _         | p < 0.05 |
| 16    | 0.99         | 0.9935         | _         | _        |
| 17    | 0.609        | 0.6043         | _         | _        |

Table 2. Table comparing the posterior probability of positive slope  $P(b_{\phi} > 0 \mid \boldsymbol{y})$  from fitting the nested AR(1) model to the different Dansgaard–Oeschger events using a second-order polynomial detrending approach. These results are compared with the probability of positive slope  $P(b > 0 \mid \boldsymbol{y})$  found by Myrvoll-Nilsen et al. (2024) and p-values obtained from Boers (2018) and Rypdal (2016).

# 4 Detecting early warning signals in AMOC fingerprint


**Figure 5.** AMOC fingerprint proxy from 1870 to 2020, similar as (Ditlevsen and Ditlevsen, 2023) using yearly averaged subpolar gyre seasurface temperature anomaly minus twice the global mean anomaly obtained from the Hadley Centre Sea Ice and Sea Surface Temperature data set (HadISST) (Rayner et al., 2003).

We now apply the time-dependent nested AR(1) model to an AMOC fingerprint similar to the one used by Ditlevsen and Ditlevsen (2023) shown in Fig. 5. This fingerprint is constructed as the sea-surface temperature (SST) anomaly in the subpolar gyre region, averaged annually, minus twice the global mean SST anomaly to compensate for the polar amplification efects under global warming. Several studies have suggested that this proxy is a suitable indicator of AMOC strength (Caesar et al., 2018; Jackson and Wood, 2020; Latif et al., 2019), especially since direct observations are only available from 2004 onward.

The use of such a proxy is therefore necessary to examine longer-term trends and detect potential early warning signals.

As the fingerprint exhibits significant drift, it must first be detrended to satisfy the zero-mean assumption of the model. In principle, this trend could be extracted using knowledge of the system's underlying physical processes, but such information may be unavailable, incomplete or inaccurate. To address this, we consider two different detrending strategies. In the first, we rely solely on statistical assumptions and remove the trend using either a linear or second-order polynomial fit. In the second approach, we incorporate physical information by including an explanatory variable in the model, following the structure described in Eq. (9). Specifically, we use integrated Central-West Greenland (iCWG) surface melt shown in Fig. 6 as a covariate. The iCWG represents the cumulative surface melt across years, based on the CWG melt stack from Trusel et al. (2018), and is used to capture the influence of freshwater forcing on AMOC stability.

Figure 6. Cumulative Central West Greenland runoff from 1871 to 2013.

For each model, we compare the posterior marginal mean estimate of the slope parameter  $b_{\phi}$  along with the posterior probability that the slope is positive. Model fit is assessed using marginal log-likelihood. The full set of results is presented in Table 3. The fitted trends and time evolutions of  $\phi(t)$  for the linear and polynomial detrending approaches are shown in Fig. 7, while the estimated response function to the iCWG forcing and associated  $\phi(t)$  evolution are shown in Fig. 8. Among the different model configurations, the version incorporating iCWG forcing provides the best fit to the data as measured by model likelihood. In all three detrending strategies, the model identifies statistically significant EWS. These results provide further evidence for the presence of EWS for the AMOC, consistent with the findings of Boers (2021) who also found statistically significant EWS using a slightly different nested AR(1) process with a window-based estimation methodology applied to a similar proxy for AMOC strength; The global mean temperature is only subtracted once in their study. Our results also corroborate those found by Ditlevsen and Ditlevsen (2023) who reported similar EWS using the same proxy but applied an AR(1) model with a window-based approach.

Figure 7. Panel (a) and (b) show AMOC fingerprint (black) with posterior marginal mean (blue) and 95% credible intervals (red) of the fitted trends. Panel (c) and (d) show the evolution in time of the correlation parameter  $\phi(t)$  (blue) used as indicator of EWS and 95% credible intervals (red) with an estimated probability of positive slope  $P(b_{\phi} > 0 \mid \boldsymbol{y})$ .

Figure 8. Panel (a) shows AMOC fingerprint (black) from 1870 to 2013 to match the time-span of the forcing data with posterior marginal mean (blue) and 95% credible intervals (red) of the estimated system's response function to forcing. Panel (b) is a plot of the evolution in time of the correlation parameter  $\phi(t)$  (blue) and 95% credible intervals (red) with an estimated probability of positive slope  $P(b_{\phi} > 0 \mid y)$ .

| Model                          | $\hat{b_\phi}$ | $P(\hat{b_{\phi}} > 0)$ | $\hat{b_{ ho}}$ | Marg. log-likelihood |
|--------------------------------|----------------|-------------------------|-----------------|----------------------|
| Nested AR(1) Linear detrending | 0.2            | 0.98                    | -0.35           | 56.49                |
| Nested AR(1) Square detrending | 0.41           | 1                       | -0.33           | 54.72                |
| Nested AR(1) Forcing response  | 0.34           | 1                       | -0.99           | 61.97                |
| AR(1) Linear detrending        | 0.145          | 0.98                    | -               | 53.46                |
| AR(1) Square detrending        | 0.278          | 0.99                    | -               | 51.68                |
| AR(1) Forcing response         | 0.19           | 0.93                    | -               | 54.13                |

**Table 3.** Summary statistics from Fig. 7 and Fig. 8 showing posterior marginal means of  $\hat{b}_{\phi}$ , probability of  $\hat{b}_{\phi}$  positive, posterior marginal means of  $\hat{b}_{\rho}$  and marginal log-likelihood for the three models used here. Results from the models introduced in Myrvoll-Nilsen et al. (2024) are also shown for comparison purposes.

# 265 5 Conclusions




This study investigates the stability of the Atlantic Meridional Overturning Circulation (AMOC) by proposing a time-dependent extension of the nested autoregressive AR(1) model introduced by Morr and Boers (2024) and Boers (2021). The primary objective of this model is to enhance the reliability of early warning signals (EWS) by minimizing false positives. This is achieved through the decomposition of the observed signal into two distinct components:  $\rho(t)$ , which captures time-dependent external variability, and  $\phi(t)$ , which reflects changes in the internal dynamics associated with system stability. By isolating these effects, the model aims to identify more accurately early signs of destabilization. Following the approach of Myrvoll-Nilsen et al. (2024), we assume a linear temporal dependence for both  $\rho(t)$  and  $\phi(t)$ , estimating their respective slope parameters within a hierarchical Bayesian framework. This statistical approach allows us to incorporate prior information and quantify the uncertainty of the EWS through the posterior distributions of the parameters. The performance of the model is first evaluated using both simulated and real data, demonstrating both high estimation accuracy and robustness against false detections of ongoing destabilization.

The methodology is applied to a proxy for the AMOC fingerprint. In order to meet stationarity assumptions, we consider various detrending techniques, including linear and second-order polynomial detrending, as well as incorporating a forcing component based on the integrated meltwater runoff from Central-West Greenland. Across all model configurations, we find statistically significant early warning signals. This is consistent with prior findings in the literature and supports the hypothesis of a possible ongoing destabilization of the AMOC.

While assuming a linear structure for  $\phi(t)$  has proven effective for detecting EWS, we emphasize that the model proposed here should not be interpreted as a comprehensive or mechanistic representation of the underlying physical processes governing the AMOC. Despite its success in identifying early signs of destabilization, the model is limited in its ability to forecast the future trajectory of the system or predict the timing of a potential tipping point. Addressing these limitations would require a more flexible modeling approach, potentially involving a nonlinear or nonparametric structure for the correlation parameters, which lies beyond the scope of the present work.

Although our analysis has focused on a specific proxy of the AMOC fingerprint, the proposed methodology is generalizable and can be adapted to study the stability of other critical climate components, such as the Greenland Ice Sheet, Arctic sea ice, or the Amazon rainforest. To facilitate wider use and reproducibility, we have extended the existing R package INLA.ews to incorporate our methodological advancements. This software provides a user-friendly interface for implementing our approach, leveraging the computational efficiency of the INLA framework for Bayesian inference.

Code and data availability. The code and data sets used for this paper is available through the R-package, INLA.ews, which can be downloaded from: github.com/eirikmn/INLA.ews (last access August 5, 2025).

*Author contributions*. All authors conceived and designed the study. LH adopted the model for a Bayesian framework and wrote the code. EMN and LH carried out the examples and analysis. All authors discussed the results and drew conclusions. All authors wrote the paper.

Competing interests. CF is a member of the editorial board of the NPG journal. LH and EMN declare that they have no conflict of interest.

Acknowledgements. CF was supported by the Institute for Basic Science (IBS), Republic of Korea, under IBS-R028-D1 and by the National Research Foundation of Korea (NRF-2022M3K3A1097082 and RS-2024-00416848). EMN has also received funding from the Norwegian Research Council (IKTPLUSS-IKT og digital innovasjon, project no. 332901). LH has also received funding from the Norwegian Research Council (Stability of the Arctic climate, project no. 314570)

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
