# Peer review of "Assessing AMOC stability using a Bayesian nested time-dependent autoregressive model"

_EGUsphere, 2025_

## Author Response (AR1)

**Review # 1**

"In this paper, a recently developed Bayesian approach (Myrvoll-Nilsen et al., 2024) to early warning signals (EWS) of tipping is applied to the time series of the Ceasar et al. (2018) AMOC fingerprint. The main result is that also this method confirms that, based on this fingerprint, the present-day AMOC is undergoing a loss of stability.

The paper is poorly written, its content is below standard for NPG and hence I recommend to reject it. The main reasons are"

Answer: We thank the referee for the comments, they will be addressed point-by-point below.

"The methodology in both sections 1 and 2 has a lot of overlap with the Myrvoll-Nilsen et al., (2024) paper."

**Answer:**

Indeed, we recognize that there is methodological overlap with Myrvoll-Nilsen et al., (2024), particularly in Sections 1 and 2. This is because our new model builds upon and extends the model presented in that paper. Moreover, we believe it is necessary to include sufficient details in order to explain the limitations of the first model, and to properly motivate how our new model addresses these limitations.

Specifically, the old model was susceptible to false positive early warning signals caused by increased external variability, instead of loss of stability. By replacing the noise component with another time-dependent AR(1) process, the new model is able to separate these signals to avoid these issues.

The manuscript now includes a direct comparison between our model and the model introduced in Myrvoll-Nilsen et al., (2024) followed by a new figure to stress out the major differences and improvements brought by our model:

**L 202-212 of the revised manuscript:**

"To illustrate the benefits of accounting for the bias introduced by correlated noise, we also test the time-dependent AR(1) model proposed by Myrvoll-Nilsen et al. (2024), which does not separate external noise autocorrelation from loss of stability. We apply this model to the same set of simulated non-tipping processes. As illustrated in Fig. 3, this model yields posterior marginal mean estimates of  $b\phi$  that correlate with the values of  $b\rho$ , rather than remaining centered around zero as expected in the absence of a true loss of stability. In contrast, our nested AR(1) model maintains stable estimates of  $b\phi$  across all values of  $b\rho$  as shown in Fig.2(a), demonstrating its robustness to external noise.

The simpler AR(1) model also exhibits a significantly higher rate of false positives, misclassifying 116 out of 500 simulations as tipping events, an approximately 400% increase compared to the nested AR(1) model. Moreover, the false positive rate increases systematically with higher values of bp, further highlighting the susceptibility of this model to bias from autocorrelated noise. In contrast, false detections in the nested AR(1) model are evenly distributed across all simulations. These results emphasize the importance of explicitly modeling the correlated noise structure when assessing stability in time series data."

Figure 3. Results of the robustness test from the Myrvoll-Nilsen et al. (2024) model applied on the same data as Fig. 3 (a). The posterior marginal mean is plotted against the true value of  $b_{\rho}$  from the correlated noise. The blue line is a linear regression on the data, showing the drift of the estimates of  $b_{\phi}$ . Blue dots are associated with a statistical significance for the EWS indicator  $b_{\phi}$  to be positive while red dots mean no statistical significance.

"Moreover, it is very poorly presented with many errors and typos (e.g. errors in equations (7) and (11)) and symbols which are only defined later in the paper (e.g. \kappa\_f in (9), F below (1), etc.)."

Answer: It is not clear which errors in Equations (7) and (11) the reviewer is referring to.

Equation (7) is the continuous-time expression of the dynamical system assumed in this paper. It is similar to the one introduced in Boettner and Boers (2022) and Morr and Boers (2024). This expression differs slightly from the more common variation of this model, where the Wiener process dW(t) is replaced by an OU process here denoted as U(t).

For Equation (11) the reviewer might refer to our inclusion of the 0.5 shift in the exponent. This is used to improve the accuracy of the integral discretization.

- The revised manuscript now defines  $k_f$  in formula (9) and only introduces and defines F below formula (7) while updating the definition of  $\lambda$  below formula (1):

L46 of the revised manuscript : "where λ is the restoring rate [...]".

L74 of the revised manuscript: "where F(t) represents the forcing [...]".

L80 of the revised manuscript : "with  $\kappa_f$  being a scaling parameter.".

"The context of the AMOC tipping problem is also poorly covered, with inappropriate references, wrong terminology (e.g. in the title, this is no 'Bayesian stability analysis' of the AMOC)."

**Answer:**

- The introduction of the AMOC tipping problem has been substantially re-written to improve its clarity and coverage of the matter and correct missing or typos in references.

**Line 14-38 of revised manuscript:**

"The Atlantic Meridional Overturning Circulation (AMOC) is a key driver of Earth's climate, responsible for the transport of heat and salt across the Atlantic Ocean (Rahmstorf, 1995). As part of the global thermohaline circulation, the AMOC plays a central role in maintaining the current climate equilibrium. It is widely believed that the AMOC is a multi-stable system, capable of existing in multiple stable modes, most notably a strong mode, which is currently dominant, and a weak or collapsed mode (Stommel, 1961; Lenton et al., 2008). This nonlinear behavior implies that the AMOC may undergo abrupt transitions between states when critical thresholds are crossed. Paleoclimate evidence supports the idea that abrupt shifts in AMOC strength have contributed to major climate events during the last glacial period, such as the Dansgaard-Oeschger events (Vettoretti and Peltier, 2016; Boers et al., 2018).

These dynamics have led to the identification of the AMOC as a potential "tipping element" in the Earth system, i.e., a subsystem that could undergo a critical transition due to anthropogenic forcing (Lenton et al., 2008). Climate models suggest that continued greenhouse gas emissions and the resulting increase in freshwater input from

Greenland Ice Sheet melt, precipitation, and river discharge could push the AMOC toward such a tipping point (Wood et al., 2019; Hawkins et al., 2011; Weijer et al., 2019). This behavior exhibits hysteresis, meaning that once a tipping threshold is passed, the AMOC may not return to its original state even if the perturbation is reversed.

Recent observational and modeling studies have intensified concerns. Although early models suggested a low probability of collapse within the 21st century (Masson-Delmotte et al., 2021), more recent simulations reveal a wider range of possible responses, raising concerns that risks might be underestimated (Gong et al., 2022). This discrepancy is partly due to model biases, notably in representing freshwater forcing and feedback (Liu et al., 2017). Evidence is also emerging from real-world observations. Studies have documented a significant weakening trend in the AMOC over the 20th century (Caesar et al., 2018) and recent statistical analyses have detected early warning signals of reduced stability (Boers, 2021; Ditlevsen and Ditlevsen, 2023). These findings suggest that the AMOC may be approaching a critical threshold.

A weakening of the AMOC would have profound and potentially irreversible consequences, including disrupting weather patterns, altering precipitation systems, and potentially triggering cascading effects on other climate components (Stouffer et al., 2006; Jackson et al., 2015; Lenton et al., 2008). In light of this, there is an urgent need to monitor the resilience of the system and improve our understanding of the processes that drive its potential loss of stability. [...]".

- Since early warning signals are related to stability, our time-dependent model provides an evolution of the stability of the system. The framework detailed in this paper thus performs Bayesian analysis of the AMOC stability. Of course, one limitation is that we assume the evolution of the autocorrelation parameter is linear. While our model is efficient at identifying whether or not there have been changes in stability, it is likely not an accurate representation of the actual evolution of the stability.

The title has been changed to "Assessing AMOC stability using a Bayesian nested time-dependent autoregressive model".

"There is no critical evaluation of the time series in Figure 4, e.g. it does not even have units on the y-axis."

**Answer:**

The cumulative surface melt across years, based on the CWG melt stack from Trusel et al. (2018) is used here to reflect the total freshwater forcing from GrIS. CWG surface melt is directly linked to GrIS runoff which is known to be one major component in the possible destabilization of the AMOC.

**The y-axis units of Fig.4 have been added:**

Figure 6. Cumulative Central West Greenland runoff from 1871 to 2013.

"The input of freshwater by the Greenland Ice Sheet has been so small over this period that a response of the AMOC is questionable."

Answer: While the freshwater by the GrIS was small prior to 1990, it has increased non-linearly since then, as shown by Trusel et al., (2018) or Horhold et al., (2023). Moreover recent studies suggest that the acceleration of GrIS melting over the past decades may have already-observable impact on the AMOC strength (Devilliers et al., 2024), (Martin and Biastoch 2023), (Castro de la Guardia et al., 2015).

"The results on the DO time series are already in Myrvoll-Nilsen et al., (2024) and cannot be understood here without consulting that paper (which data, etc.?)."

**Answer:**

The demonstration of our model on DO time series is meant to serve as a benchmark to compare with other approaches on a real data example, in addition to the other two tests assessing the accuracy of our model fitted onto simulated nested AR(1) processes and simulated tipping processes respectively.

The results from the approach of Myrvoll-Nilsen et al.,(2024) are shown in Table 2, as are the results of Rypdal (2016) and Boers (2018). While there is no ground truth regarding which (if any) DO events are bifurcation-induced and should therefore exhibit EWS, they present a classical example of real tipping points and have been extensively

studied in the literature. They therefore provide a natural and informative case study for comparing against other approaches. Our results appear to more or less corroborate previous results.

The revised manuscript now includes an additional plot of NGRIP d18O record data used to study the DO events followed by a more detailed description of this experiment:

**Line 216-238 of revised manuscript:**

Figure 4. The 17 most recent Dansgaard-Oeschger (DO) events (vertical black lines) in the NGRIP  $\delta^{18}$ O record plotted against the GICC05 chronology. Early warning signals are estimated by fitting the model to the Greenland stadial periods (black segments) of the data preceding each DO event.

"Finally, we evaluate the performance of our model on real-world data associated with well-studied critical transitions. By comparing results, we can assess how much our model agrees or disagrees with existing approaches. Specifically, we will use our model to analyze abrupt Greenland warmings known as Dansgaard-Oeschger (DO) events (Dansgaard et al., 1993; Johnsen et al., 1992), which represent rapid climate fluctuations that occurred during the last glacial period, where the temperature over Greenland and the North Atlantic region increased by up to 16.5°C within a few decades (Kindler et al., 2014). DO events are often considered the archetypal example of tipping point crossings in the climate. As such, they present a natural benchmark for evaluating different EWS approaches.

For our analysis, we pair the δ18O proxy data from the Northern Greenland Ice Core Project (NGRIP) (North Greenland Ice Core Project members, 2004; Gkinis et al., 2014; Ruth et al., 2003) with the corresponding age provided by the Greenland Ice Core Project 2005 (GICC05) (Vinther et al., 2006; Rasmussen et al., 2006; Andersen et al., 2006; Svensson et al., 2008). The data is available at

https://www.iceandclimate.nbi.ku.dk/data (last accessed: August 1, 2025). The model is fitted to segments preceding the 17 most recent DO event. The selected segments are highlighted in Fig. 4.

Whether or not DO events are induced solely by noise, or if they are indeed approaching a bifurcation point, is currently debated (Ditlevsen et al., 2007; Hummel et al., 2024). There is therefore no ground truth as to which, if any, DO event should exhibit EWS. However, several studies report a detection of EWS before some of the first 17 DO events (Rypdal, 2016; Boers, 2018). We compare the results of our model with these studies using a setup similar to Myrvoll-Nilsen et al. (2024) by using a second-order polynomial detrending of the data and considering  $P(b\phi > 0 \mid y) > 0.95$  as a detection of EWS. This comparison is illustrated in Table 2 and shows that our model suggests, similarly to Myrvoll-Nilsen et al. (2024), that some specific event shows signs of critical slowing down in line with the results of Boers (2018) and Rypdal (2016). Specifically, Table 2 shows that our results corroborate the five EWS found by Myrvoll-Nilsen et al. (2024) while identifying one more EWS for the 13th event. Moreover, these results corroborate the EWS found for the 11th event by Boers (2018) and the 5th and 9th events by Rypdal (2016), our results also show EWS for the 2nd and 13th events similarly to these two studies. ".

"The results for the AMOC fingerprint are in terms of application the only new results. These are poorly described and one would at least expect a comparison with other methods."

Answer: In terms of application, this paper focuses indeed solely on the AMOC. However, the paper also introduces new methodology using a time-dependent nested AR(1) process that accounts for biases arising from structured external variability, addressing a core limitation of Myrvoll-Nilsen (2024). The methodology introduced is robust and general enough to be applied to other climate systems prone to tipping, therefore it constitutes another greatly significant result of this paper.

We do provide a comparison in Table 3 with the model introduced in Myrvoll-Nilsen (2024) and show that using a forcing response as detrending our new model is able to detect EWS while the previous one is not. Furthermore we mention at the end of Section 3 that the results of this study are consistent with the ones of Boers (2021) and Ditlevsen and Ditlevsen (2023).

A more detailed discussion on the comparison with Boers (2021) and Ditlevsen and Ditlevsen (2023) has been added:

Line 259-264 of revised manuscript:

"These results provide further evidence for the presence of EWS for the AMOC, consistent with the findings of Boers (2021) who also found statistically significant EWS using a slightly different nested AR(1) process with a window-based estimation methodology applied to a similar proxy for AMOC strength; The global mean temperature is only subtracted once in their study. Our results also corroborate those found by Ditlevsen and Ditlevsen (2023) who reported similar EWS using the same proxy but applied an AR(1) model with a window-based approach.".

"I would recommend to the authors to add the AMOC fingerprint example in the Myrvoll-Nilsen et al., (2024) paper."

Answer: The study presented here indeed builds upon the hierarchical Bayesian framework developed in Myrvoll-Nilsen et al., (2024). However, the methodology differs significantly from that paper. As stated in the introduction, regular AR(1) processes (including the time-dependent version presented in Myrvoll-Nilsen et al., (2024)) do not account for structured external variability. On the other hand, this new nested AR(1) process can account for such biases.

The updated manuscript puts a greater emphasis on the important differences between these two methodologies. The changes brought in the revised manuscript are shown in the first answer of this letter.

**Review # 2**

**"General comments:**

The Atlantic Meridional Overturning Circulation (AMOC) is a key climate tipping element. The author claims that they proposed a robust and general statistical model based on a second-order autoregressive process featuring time-dependent parameters. These parameters separately account for the statistical changes arising from increased external variability and destabilization. By applying the model to an AMOC fingerprint proxy, the author detected statistically significant early warning signals (EWS) of declining AMOC stability and an approaching tipping point. This manuscript is well written, with clear descriptions, an appropriate length, and high-quality figures. The analysis is appropriate. It represents a valuable contribution to the field. I recommend accepting it after some minor revisions."

Answer: We thank the referee for the constructive review. The comments will be addressed point-by-point below.

"Minor comments:

1. Some sentences in the manuscript are too long in length; consider revising them to improve readability. (e.g. Line 5-7)"

**Answer:**

- Line 5-7 have been changed to:

"To assess these changes in statistical behavior we propose a robust and general statistical model based on a second-order autoregressive process with time-dependent parameters.

This allows for the statistical changes from increased external variability and destabilization to be accounted for separately.".

- Substantial parts of the manuscript have been re-written to improve readability.
- "2. Line 30, add missing letter "s" in "called early-warning signals (EWSs)""

**Answer:**

- Line 30 have been changed to :

Line 40 in revised manuscript: "This phenomenon, called early-warning signal (EWS) [...]".

"3. Line 148, are 500 independent simulations enough?"

Answer: For each test, Root Mean Squared error (RMSE) has been plotted against the number of independent simulations. Results show that the RMSE values tend to converge around 300 independent simulations. Therefore we believe that 500 independent simulations is enough.

"4. The subscripts in the captions of Fig. 2(a) and (c) appear to be incorrect, please check."

**Answer:**

The caption of Fig. 2(a) and (c) has been corrected is now:

"In panels (a) and (c) blue dots are associated with a statistical significance for the EWS indicator  $b\phi$  to be positive  $P(b\phi > 0 | y) > 0.95$  while red dots mean no statistical significance  $P(b\phi > 0 | y) < 0.95$ ".

"5. Line 191-192, the citation style for references needs to be modified. It is recommended to consolidate them into a single bracket."

Answer:

Phrasing and citations style in line 191-192 have been changed to :

Line 229-232 of revised manuscript:

"Whether or not DO events are induced solely by noise, or if they are indeed approaching a bifurcation point, is currently debated (Ditlevsen et al., 2007; Hummel et al., 2024).

There is therefore no ground truth as to which, if any, DO event should exhibit EWS. However, several studies report a detection of EWS before some of the first 17 DO events (Rypdal, 2016; Boers, 2018).".

"6. There is an unexplained discrepancy: the abstract mentions "a second-order autoregressive process with time-dependent parameter", but Section 3.1 exclusively discusses "time-dependent nested AR(1) model". What's the relationship between the AR(2) model and the time-dependent nested AR(1) model? This should be clarified."

Answer: The denomination "nested time-dependent AR(1) model" or "nested AR(1) model" are now replacing all mentions of "AR(2) model" to improve clarity.

**Review # 3**

"This study proposed a nested time-dependent autoregressive model to investigate the early warning signals (EWS) of AMOC, following the work of Myrvoll-Nilsen et al. (2024). The proposed model was tested to find the EWS. The results showed that it can indeed better find warning signals. But some issues need to be addressed before considering for publication."

Answer: We appreciate this helpful review. The comments will be addressed point-by-point below.

"1)The math symbols are different in the formula and the text. For example, line 49 and formula (4). Line 65 and formula (9). Besides, what does the k\_f denote in formula (9)?"

**Answer:**

- The notations have been updated in line 49 and 65 to :

L60 of the revised manuscript : "[...] the white noise process  $\varepsilon_t$  is replaced by an AR(1) process".

L78 of the revised manuscript : "[...], v(t), is expressed by ".

- The revised manuscript now defines k f in formula (9):

L80 of the revised manuscript : "with  $\kappa_f$  being a scaling parameter.".

"2)Line 193. Please explain why 0.95 is used as a detection of EWS. In Table 2, which events are real occurrences. If this information is provided, it may be easier to understand the disadvantage of the proposed model."

**Answer:**

The reason why 0.95 was chosen is because that is the conventional threshold for statistical significance. It is ultimately arbitrary, and we will add a comment on this in the revised manuscript on why this threshold was chosen.

Table 2 presents the results of testing our model on real data, from the Dansgaard-Oeschger (DO) events. It is debated which of these events (if any) are actually caused by bifurcations, hence there is no known truth to compare to. However, the model appears to corroborate previous results, lending credence to our model relative to existing approaches.

"3)Line 196. 2 should be Table 2? Line 214. 1971 should be 1871?"

**Answer:**

- Lines 196 have been changed to:

L235 of the revised manuscript: "Specifically, Table 2 shows [...]".

- Lines 214 have been changed to:

L253 of the revised manuscript: "Cumulative Central West Greenland runoff from 1871 to 2013.".

"4)In this study, different detrending strategies were applied. Table 3, Figures 5 and 6 show that the results seem to depend on the detrending strategies. Then, do you think which strategy should be used in finding the EWS?"

Answer: While the results might differ between the detrending strategies, they all agree on the presence of early warning signals. Generally, one should select the detrending strategy that is most appropriate for the problem at hand. If forcing data is available, and physically meaningful, it would often be beneficial to incorporate that into the model, over the more basic detrending strategies.